# Communication Hierarchy-aware Graph Engine for Distributed Model Training

## ABSTRACT

Efficient processing of large-scale graphs with billions to trillions of edges is essential for training graph-based large language models (LLMs) in web-scale systems. The increasing complexity and size of these models create significant communication challenges due to the extensive message exchanges required across distributed nodes. Current graph engines struggle to effectively scale across hundreds of computing nodes because they often overlook variations in communication costs within the interconnection hierarchy. To address this challenge, we introduce TuComm, a communication hierarchy-aware engine specifically designed to optimize distributed training of graph-based LLMs. By leveraging hierarchical network topology, TuComm dynamically aggregates and transfers messages, fully accounting for the underlying communication domains, thereby enhancing the efficiency of distributed model training across large-scale systems. We implemented TuComm on top of the message passing interface (MPI), incorporating innovations such as dynamic buffer expansion and active buffer switching to enhance scalability. Evaluations conducted on synthetic and real-world datasets, utilizing up to 79,024 nodes and over 1.2 million processor cores, demonstrate that TuComm surpasses leading graph-parallel systems and state-of-the-art counterparts in both throughput and scalability. Moreover, we have deployed TuComm on a production supercomputer, where it consistently outperforms top solutions on the Graph500 list. These results highlight TuComm's potential to significantly enhance the efficiency of distributed large-scale graph-based LLM training by optimizing communication among distributed systems, making it an invaluable communication engine for web-scale model training.

## CCS CONCEPTS

• **Computing methodologies → Distributed algorithms**; Massively parallel algorithms.

## KEYWORDS

communication hierarchy, message aggregation, communication domain, graph-LLM training, Graph500

## 1 INTRODUCTION

Recent advances in distributed model training, particularly for graph-based large language models (LLMs) [6, 8, 35, 63, 74, 77], have increasingly relied on efficient graph processing techniques. As models grow larger and more complex, the amount of data they require has expanded significantly, often in the form of massive graphs representing web-scale information [33, 62, 69], social networks [56, 74, 85], or structured data [3, 22, 23, 48, 86, 87]. These graphs, comprising hundreds of billions to trillions of vertices and edges [20, 23], present unique challenges for distributed computing systems. Training models with such vast datasets demands the use of parallel and distributed infrastructures that can efficiently process and communicate between thousands of computing

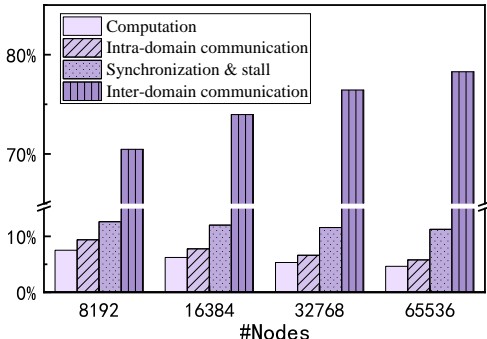

**Figure 1: The breakdown for BFS execution time.**

nodes (CNs[1]). This paradigm shift in distributed model training which characterized by the need to process ever-expanding graphs underscores the importance of scalable graph engines capable of handling this computational complexity. Large-scale distributed systems, such as supercomputers with thousands of CNs, play a pivotal role in managing these enormous graphs. As we approach the era of 100 trillion parameters [45], a distributed computing system of this magnitude typically comprises hundreds even thousands of CNs. For example, TaihuLight [48, 87] successfully processes the Sogou graph [84] with over 271.8 billion vertices and 12.3 trillion edges, while the Fugaku supercomputer handles Kronecker graphs with 70.4 trillion edges across 152,064 nodes [32]. These achievements illustrate the potential for distributed systems to scale to the demands of exascale graph processing [22, 23, 48].

In the realm of distributed training for graph-based LLMs, model training time can be divided into three aspects [20, 21]: 1) computation; 2) communication; 3) synchronization&stall. Moreover, communication can be further classified as intra- and inter-domain communications. Computation time is accumulated from all the computing nodes during computation. Intra-domain communication is communicating time between the CNs attached to the same routing cell (RC[2]), and inter-domain communication is communicating time between two connected RCs in two different communication domains. Figure 1 offers a breakdown of large-scale graphs running on thousands of CNs with hierarchical communication domains. We can observe that communication becomes a bottleneck in graph computing with thousands of CNs. Further, we can also observe the occupation of inter-domain communication increases as the number of CNs increases, i.e., the percentage of inter-domain communication increases from ca. 70% to ca. 80% as the number of CNs increases from 8192 to 65536. Such a trend underscores the significant variability of inter-domain communication costs across different domains. Therefore, large-scale graph-based LLMs training rely heavily on distributed communication optimization for their success [7, 23–26, 48, 49, 86, 87]. As such, various graph communication strategies have been proposed

---

[1]A CN may have one or multiple CPUs or accelerators[22, 75].

[2] It is responsible for connecting CNs, e.g., routers [20].

[1, 16, 17, 25, 38, 39, 53, 55, 59, 60, 67, 87]. Indeed, all parallel graph processing systems utilize some form of graph communication optimization to leverage architectural advantages [22, 23, 48, 87] for better performance. This emphasizes the need for sophisticated communication engines that can address the unique challenges posed by large-scale graph training in LLMs, ensuring that computation, communication, and synchronization are effectively balanced to enhance performance and aligns with the theme of distributed model training, connecting the challenges in communication with the need to optimize the training of LLMs.

Unfortunately, current graph processing engines always assume consistent communication overhead between any two CNs. This is because the existing graph engines are cluster-based systems with only tens of CNs, which makes them inadequate for training LLMs with large graphs containing trillions of edges and vertices across thousands of distributed CNs. Even in configurations involving hundreds of CNs organized into hierarchical communication domains where groups of CNs are interconnected via dedicated networks, the variability in communication overhead across different RCs remains substantial. This variability can be observed in the analysis presented in Figure 1.

To effectively manage communication in large-scale distributed LLM training, it is crucial to develop a robust message library that leverages the hierarchical communication domains present in high-performance computing (HPC) systems, such as supercomputers. For instance, the state-of-the-art Active Messages Library (AML) [27] supports each source node to aggregate messages that are targeted to the same domain. However, the communication cost of AML is still overwhelming when processing trillion-scale graphs on exascale clusters, which severely affects the graph searching performance and the reasons are listed as follows. (i) AML aggregates messages only at the source nodes, and ignores opportunities of aggregating messages in higher level communication domains; and (ii) AML only supports static buffer management, which not only lowers the graph processing performance but also is vulnerable to a buffer overflow when aggregating large numbers of messages. That is because there are numerous inter-domain (i.e., across RNs) communications for AML running in large-scale clusters and inter-domain communication is more expensive than intra-domain communications, i.e., by up to orders of magnitude [23]. An advanced MST [19] based on TianheGraph [22] is proposed for aggressive aggregation, but it lacks awareness of communication hierarchies. Furthermore, both MST and AML only support static buffer management. As such, it is essential to build an efficient message transfer engine, by taking advantage of hierarchical communication domains within large-scale HPC systems with efficient buffer configuration.

In AML-like communication libraries, inter-domain message transfers are not only frequent but also significantly more expensive than intra-domain communications. This cost disparity often arises from the physical network architecture of large-scale HPC systems. Typically, intra-domain communication, where nodes within the same domain exchange messages, is relatively fast due to shorter communication paths and reduced latency, often taking as little as $0.1\mu s$. However, when communication must occur between different domains, inter-domain transfers introduce much higher latency, sometimes as much as $1\mu s$ or more. This is because inter-domain communication often traverses additional network layers or even entirely different network segments, increasing the time it takes for messages to reach their destination.

TuComm[3] offers application programming interfaces (APIs) for fundamental graph processing operations, including breadth-first-search (BFS), single source shortest paths (SSSP), connected component (CC) [18], betweenness centrality (BC) [9, 78], page ranking (PR) [82], and community detection with label propagation (CDLP) [50]. It has been deployed on the production environment of the Tianhe-Exa supercomputer [51] and has supported a diverse range of graph applications.

We evaluate TuComm by applying it to representative graph operations, including BFS, SSSP, CC, BC, PR and CDLP. Our evaluations use three famous supercomputers with varying scales, using up to 79,024 nodes and over 1.2 million processor cores. We show that TuComm consistently outperforms state-of-the-art AML-like libraries and graph systems [1, 2, 14, 37, 39, 40, 43, 59, 60, 87] on different graph scales and hardware setups. Specifically, TuComm achieves 162,494 and 23,021 giga-traversed edges per second (GTEPS), respectively, for BFS and SSSP according to the Graph500 specification [32]. These results are translated to a 1.19× and 1.5× improvement for BFS and SSSP over the top-ranked system on the Graph500 ranking (Nov. 2023). We also test TuComm on real-world graphs, for which it outperforms three state-of-the-art graph processing engines, GraphScope [14], Gluon [11] and GraphCube [20], with a speedup of up to 27.34×.

This paper makes the following contributions:

- It offers analytical formulas to model communication cost of large-scale HPC systems with hierarchical communication topology (Sec. 4);
- It proposes an interconnection hierarchy-aware message aggregation method designed to minimize cross-domain communications, thereby enhancing efficiency in distributed graph-based LLM training environments (Sec. 5).

## 2 BACKGROUND

### 2.1 Communication Hierarchies of HPC

Large-scale HPC systems often implement a hierarchical communication topology [12, 46, 75, 80] and use RCs to link different CNs. We use Tianhe-Exa [51], the main evaluation system of this work, to highlight the differences in the communication latency at different levels of the topology. Tianhe-Exa is an upgrade of Tianhe-2A [68]. The interconnection architecture of Tianhe-Exa is similar to that of many of today's large-scale HPC systems [75, 80].

Similar to other large-scale HPC systems [4, 12, 20, 22, 23, 48, 61, 70, 71, 81], Tianhe-Exa implements a hierarchical interconnect topology with varying communication latencies. CPUs are connected via an onboard network mesh with nanosecond-level access latency at the CMU level. Blades within a shelf use an in-cabinet switch with microsecond-level data transfer latency. CMUs are linked across shelves using a customized networking router with sub-millisecond-level latency. Racks are connected by top-level switches, providing millisecond-level communication latency. Communication among nodes across shelves is about twice as slow as

---

[3]Code available at *https://anonymous.4open.science/r/graph-com-2499/README.md*

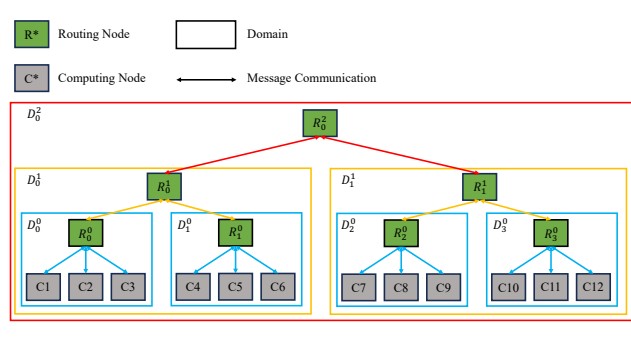

**Figure 2: A model of the hierarchical communication topology of large-scale HPC system with 3-level communication domain.**

within the same shelf, and across racks, the latency increases to around 15 times that within a shelf [5, 20, 46, 61, 75].

## 2.2 Graph Processing & Graph500

Graph processing algorithms are usually communication-intensive [13, 28, 36, 52, 54, 65, 66, 79, 83, 87] in that huge numbers of *small* messages are transferred through the interconnection network. Graph500 [32] is the de facto standard for benchmarking and ranking the graph processing performance of large-scale HPC systems (e.g., supercomputers), using the TEPS (traversed edges per second) metric. Currently, Graph500 has two separate ranking lists respectively measuring the BFS and SSSP performance [32].

Following Graph500 [32], we report the graph processing performance using the GTEPS metric by counting the number of TEPS. This is a *higher-is-better* metric. Note that Graph500 is often used to evaluate HPC system performance for data-intensive workloads [20, 22, 47, 48, 57, 71, 73].

## 3 PRELIMINARIES

### 3.1 Definitions

DEFINITION 3.1. **Communication Domain**. A communication domain, $D$, is a set of CNs that are attached to the same RC, which can be represented as $D = \{R, C_1, C_2, \cdots, C_n\}$, where $n$ is the total number of CNs (denoted as $Cs$) in $D$, and $R$ is the RC that all the CNs are connected with. The communication domain can be further classified into the leaf domain and high-level domain, respectively. A leaf domain is attached to a leaf RC that is directly responsible for a set of CNs. For example, in Figure 2, leaf domain $D_0^0 = \{R_0^0, C_1, C_2, C_3\}$ because $C_1, C_2, C_3$ are attached to $R_0^0$. While the parent domain (i.e., high-level domain) includes a set of CNs attached to the same high-level RC. For example, $D_0^1 = \{R_0^1, C_1, C_2, C_3, C_4, C_5, C_6\}$, since all the CNs are attached to $R_0^1$.

DEFINITION 3.2. **Communication Cost.** The communication cost (ComCost) between two nodes, e.g., $Ci$ and $Cj$, can be defined as $ComCost(Ci, Cj) = ComCost_i^{intra} + ComCost_j^{intra} + \sum_{l=1}^{h} ComCost_{i_{(l-1,l)}}^{inter} + \sum_{l=1}^{h} ComCost_{j_{(l-1,l)}}^{inter}$, where $ComCost_i^{intra}$ is the intra-domain communication cost among the RC that $Ci$ belongs to, $ComCost_{i_{(l-1,l)}}^{inter}$ is the inter-domain communication cost between the $(l-1)-th$ level

communication domain $D^{l-1}$ and the $l-th$ level communication domain $D^l$ that $C_i$ belongs to, and $h$ is the lowest domain level where both $Ci$ and $Cj$ are located at.

Figure 2 gives an example of a large-scale HPC system with a 3-level communication domain. In this example, we have 12 CNs and 7 RCs, which are separated into 7 communication domains, i.e., $D_0^0, D_1^0, D_2^0, D_3^0, D_0^1, D_1^1, D_0^2$, where $D_i^j$ denotes the $j$-th level domain and we call $D_*^0$ the leaf domain. Each communication domain $D_i^j$ is associated with one RC, denoted as $R_i^j$, so the domains can be represented as $D_0^0 = \{R_0^0, C1, C2, C3\}$, $D_1^0 = \{R_1^0, C4, C5, C6\}$, and $D_0^1 = \{R_0^1, C1, C2, C3, C4, C5, C6\}$. We can observe that $C1$ and $C4$ are in different leaf domains, i.e., $D_0^0$ and $D_1^0$, but they are in the same high-level communication domain $D_0^1$, i.e., the $1-st$ level communication domain. In practice, the communication costs differ a lot between intra- and inter-domain communications. For example, the intra-domain communication cost can be 1 unit[4], i.e., $ComCost_*^{intra} = 1 \mathcal{U}$, while the inter-domain communication cost between $D_*^0$ and $D_*^1$ can be 10 $\mathcal{U}$, i.e., $ComCost_{*(0,1)}^{inter} = 10 \mathcal{U}$.

### 3.2 Problem Formulation

Given a graph $\mathbf{G} = (V, E)$ distributed into a target large-scale HPC system (a.k.a., **Exa**). A communication engine aims to exchange messages among CNs by minimizing the total message communication costs from all vertices, which can be formulated as follows.

$$\min \sum_{i=1}^{N} \sum_{j=1}^{N} ComCost(v_i, v_j), \quad (1)$$

$$\text{subject to } v_i, v_j \in \mathbf{Exa}.\text{CNs}.$$

where $N$ is the total vertices in $\mathbf{G}$, and **Exa**.CNs refers to the computing node set belonging to **Exa**. $ComCost(v_i, v_j)$ is the message communication cost between $v_i$ and $v_j$ which are distributed into CNs equipped in the **Exa**.

## 4 OVERVIEW OF TUCOMM

TuComm is designed to optimize large-scale graph processing on thousands or more computing nodes. It explicitly considers the interconnection hierarchy during graph communication. This is accomplished by using analytical models to aggressively perform interconnection hierarchy-aware message aggregation where messages are gathered within the domain at each level and scattered in the target domains, aiming to reduce the communication latency by transmitting expensive inter-domain into cheap intra-domain communication. This is completely different from traditional communication mechanisms, such as the message aggregation of AML and MST, where they first gather messages across domains and then scatter them within a domain [5, 20, 22, 27, 29, 61].

**Implementation.** We have implemented TuComm as a library in around 20K lines of C/C++ code. It provides APIs for common graph operations, including those evaluated in this work.

### 4.1 Preliminaries

We approximate the communication delay (i.e., communication cost), $d_{i,j}$, of two computation nodes, $n_i$ and $n_j$, as:

---

[4] A $\mathcal{U}$ maybe one microsecond, millisecond or second depending on the target system.

$$\begin{cases} d_{i,j} = d_0^i + d_0^j + \sum_{l=1}^{h} d_l^h \\ d_l^h = d_0^k + \sum_{h=k+1}^{H} d_h \end{cases} \quad (2)$$

where $d_0^i$ (or $d_0^j$) is the communication latency of $Ci$ (or $Cj$) within the local domain, $d_h$ is the latency at a high-level domain (if cross-domain communication is required between $Ci$ and $Cj$), and $H$ is the top-level of communication required. The latency of communication at each interconnection level is affected by the number of hops needed to transfer messages among computing nodes [5, 61].

### 4.2 Modeling Communication Latency

Solving Eq. 1 is in NP. Its NP-hardness could be validated by reduction from the set partition problem [15, 30, 42]. Thus, our ultimate objective is to approximate the accumulative communication delay between any two CNs as shown in Eq. 3. This can be formulated as:

$$\min \sum_{i=1}^{N} \sum_{j=1}^{N} ComCost(v_i, v_j) = d_0 + d_l^h \quad (3)$$

According to the definitions in the subsection 3.1, $d_0$ and $d_l^h$ are the intra-domain and inter-domain communication costs via RCs, respectively. They can be further formulated as:

$$\begin{cases} d_0 = \sum_{i=1}^{N} ComCost_i^{intra} + \sum_{j=1}^{N} ComCost_j^{intra} \\ d_l^h = \sum_{l=1}^{h} \left( \sum_{i=1}^{N} ComCost_{i(l-1,l)}^{inter} + \sum_{j=1}^{N} ComCost_{j(l-1,l)}^{inter} \right) \end{cases} \quad (4)$$

Clearly, the $d_l^h$, i.e., cross-domain communication takes a majority of the accumulative communication costs and dominates the communication cost for large-scale graph processing on large-scale HPC systems according to Eq. 4. That is because (i) inter-domain communication delay is orders of magnitude higher than that of intra-domain and (ii) there are a vast number of cross-domain (i.e., inter-domain) communications in large-scale graph processing within hierarchical communication domains [20, 23, 48, 71].

To mitigate cross-domain communication, we present TuComm, an aggressive message aggregation strategy designed to maximize the benefits of hierarchical communication topology by substituting costly inter-domain communication with cost-effective intra-domain communication. In order to facilitate message aggregation, we equip TuComm with flexible buffer management including active buffer switching and dynamic buffer expansion for further advancing large-scale graph processing.

## 5 HIERARCHY-AWARE AGGREGATION

Huge performance gap between intra-/inter-domain communication motivates us to significantly reduce the number of cross-domain messages. Specifically, TuComm proposes an interconnection hierarchy-aware message aggregation mechanism where messages are gathered in the source domains and scattered in the target domains.

In practice, AML refines a (global) communication into two sub-communications: an inter-domain communication (comm_inter) which happens between two nodes in the same row of different domains, followed by an intra-domain communication (comm_intra) which happens between two nodes in the same column (i.e., inside the same domain). Figure 3(a) illustrates the global communication of AML. A message from node 4 to node 2 will first be transferred to the target domain, i.e., from node 4 to node 0 in the same row

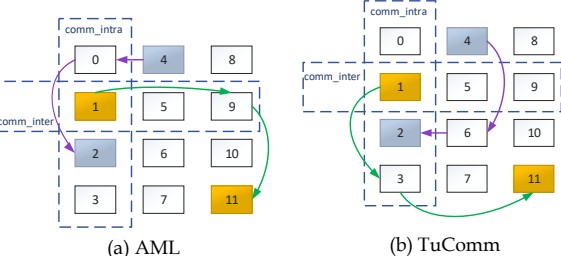

(a) AML  (b) TuComm

**Figure 3: Comparison of global communication in AML and TuComm. TuComm is more aggressive in message aggregation in that it aggregates messages in the source domains.**

(comm_inter), and then transferred from node 0 to node 2 in the same column (comm_intra). Similarly, a message from node 1 to node 11 will first be transferred from node 1 to node 9 (comm_inter), and then transferred from node 9 to node 11 (comm_intra). Although AML's communication paradigm (comm_inter followed by comm_intra) facilitates its per-node message aggregation, it prevents aggregation of messages from different nodes in the same domain. This limitation makes it inefficient for large-scale graph processing on large-scale systems.

To improve communication efficiency, TuComm leverages the topology information of interconnection networks to perform domain-level (rather than node-level) message aggregation, where messages destined for the same target domain are aggregated in the source domain before transmission. To achieve this, messages have to be transferred in a different way from AML. Figure 3(b) shows the message transmission based on TuComm: a message from node 4 to node 2 will first be transferred to the source domain, i.e., from node 4 to node 6 in the same column (comm_intra), and then from node 6 to node 2 in the same row (comm_inter); and a message from node 1 to node 11 will first be transferred from node 1 to node 3 (comm_intra), and then transferred from node 3 to node 11 (comm_inter).

Accordingly, TuComm realizes domain-level message aggregation through the following steps. First, the monitor node of the source domain gathers small messages within the same target domain (intra-domain communication) and packs them into a long message. Second, the monitor node in the source domain transmits the aggregated long message to the monitor node in the target domain (inter-domain communication). Third, the monitor node in the target domain scatters the messages to their target nodes (intra-domain communication).

## 6 TUCOMM IMPLEMENTATION

We have implemented TuComm on top of MPI (version 3.2.1). TuComm provides the standard MPI interface for message transmission with support for different type and size of messages.

### 6.1 Hierarchy-aware Message Aggregation

Algorithm 1 outlines communication hierarchy-aware message aggregation for huge messages when processing large-scale graphs. The algorithm operates on a list of allocated computing nodes, denoted by $\mathbb{N}$, which contains the node IDs. TuComm considers both

**Algorithm 1:** Communication hierarchy-aware Aggregation

**Input:** Sorted vertex list, $V$, a list of computing nodes, $\mathbb{N}$, clustering distance threshold, $h$

// Build communication hierarchy and return the level of communication hierarchy

1   Organize computing nodes into a communication hierarchy with hierarchical communication domains according to the communication topology of target systems;

2   L← $hierarchy(\mathbb{N})$

3   i = L

4   **while** $i \geq 0$ **do**

5      grouping the communication domains as $\mathcal{D}^{(L-i)}$ from bottom to up according to Figure 2;

6      i=i-1;

7   **end**

8   MPI initialization

9   Create buffers at receivers and senders, respectively

10   Vote monitors for communication domains

11   Wait for msgs of intra/inter domains at monitors

12   **while** $!empty(V)$ **do**

13      $v \leftarrow V.dequeue()$

14      **Aggregation**($v$)

15   **end**

16   **Function** Aggregation(*vertex v*)

17      $\mathfrak{C}_v$=[]

18      **for** *(i=0; i<L; i++)* **do**

         // Calling Algorithm 2.

19        $\mathfrak{C}_v^i \leftarrow$ **Gathering**($v$, $i$)

20        Scattering $\mathfrak{C}_v^i$ to intra-domain target nodes

21      **end**

22      **while** $!empty(\mathfrak{C}_v)$ **do**

23        i=0;

24        **while** $i < L$ **do**

25          do $\mathfrak{C}_{max} \leftarrow \max(\mathfrak{C}_v)$

26          Scattering messages in $\mathfrak{C}_{max}$ among nodes attached to $D^{(i)}$;

           i=i+1;

27        **end**

28

29      **end**

30   **end**

---

**Algorithm 2:** MPI-based message gathering

**Input:** vertex $s$ and specified communication hierarchy $h$

**Output:** $\mathfrak{C}$: gathering small messages at communication domain for $v_0$

1   **Function** Gathering($v_0$, $h$)

2      **while** *msg received at monitor* **do**

3        **if** *msg from the same intra-domain* **then**

4          buffer[i] ← msg; // According to targets

5          **if** *buffer[i].size ≥ threshold* **then**

6            Aggregate messages in buffer[i];

7            barrier;

8            Switch active/reserved buffers.

9            Let the remote monitor corresponding to the switched buffer call TuComm _register_handler;

10          **end**

11        **end**

12      **end**

13      return $\mathfrak{C}$

14   **end**

---

the locality of graph and communication differences across communication hierarchies together to reduce communication overheads and effectively utilize the bandwidth.

TuComm (Algorithm 1) first builds a communication topology for target large-scale HPC systems, grouping computing nodes into communication domains according to the interconnection hierarchy of the target HPC systems and getting the total levels of communication hierarchy (lines 1~7). Following, the MPI library is initialized (Line 8). Then the receiver and sender nodes create recv/send buffers, and each communication domain selects one monitor, which serves as the domain's gateway waiting for cross-domain communication (Lines 9~11).

The aggregation function selects the vertex in **V** with the highest degree and utilizes the given vertex to group small messages into a cluster, $\mathfrak{C}$, of which the messages are assigned to computing nodes recursively to adapt to the target communication hierarchies. This method prioritizes node placement within the same communication domain or domains at the same level. To aggregate messages based on the communication domain, we specify a communication level, denoted as $L$. We then generate a list of $L$ clusters, $\mathfrak{C}_v = \mathfrak{C}_v^1, \mathfrak{C}_v^2, \ldots, \mathfrak{C}_v^L$, where each cluster $\mathfrak{C}_v^i$ corresponds to a communication level of $i$ ($1 \leq i \leq L$) from the highest-degree

vertex $v$ that has not been processed yet, to scatter messages into intra-domain nodes (lines 16~20). TuComm recursively starts from the lowest level of the communication hierarchy available and scatters messages to a higher level than the current level, if the communication domain cannot hold all messages in $\mathfrak{C}$.

## 6.2 Message Transfer

The pseudo-code of the basic implementation of message transfer is summarized in Algorithm 2. After a monitor is configured at each communication domain, it serves as the domain's gateway waiting for inter-domain communication. If the monitor receives a message from within its domain that needs to be transmitted to another domain, then it adds the message to a send buffer according to the target domain (Lines 2~4). Once the buffer size reaches the threshold, messages targeted to the same domain will be gathered as an aggregate message, which will be transmitted to the monitor node in the target domain (*i.e.*, inter-domain communication at Lines 5~7). Once the messages in the buffer are transmitted, the monitor switches the empty buffer with a waiting buffer for another domain (Line 8). The remote monitor corresponding to the switched buffer then calls TuComm _register_handler() to register the handler function for the new target domain (Line 9).

## 6.3 TuComm-based BFS on Tianhe-Exa

This subsection briefly introduces how we leverage TuComm to realize the BFS test of Graph500 on Tianhe series supercomputers. Other TuComm-based graph operators including SSSP, CC, BC, PR and CDLP are realized similarly to BFS and thus are omitted due to space limitation.

Kronecker-generated graphs are skewed in vertex degree distribution: a small proportion of vertices have very high degrees. High-degree vertices (a.k.a., heavy vertices) need buffering, because

**Table 1: Hardware systems used in our evaluation**

| System | CPU | Max #Comp. Nodes | RAM per node | Top-level bandwidth |
|---|---|---|---|---|
| Tianhe-Exa | 16-core FT-2000 ARMv8 CPU @ 2.2 GHz | 79,024 | 16G | 200Gbps |
| Intel Cluster | 12-core Intel Xeon CPU @ 2.93 GHz | 512 | 64G | 160Gbps |
| WuzhenLight | 64-core HG2 7285H (AMD x86 ISA) CPU @ 2.5 GHz | 1,024 | 256G | 100Gbps |

the workload and communication traffic are higher for heavy vertices than for low-degree vertices. Therefore, we sort all vertices with buffering in the preprocessing stage, assigning ID 0 to the vertex with the highest degree. We maintain a mapping for each vertex between its new ID (according to its degree) and original ID.

To adapt graph processing to the network topology, we refactorize graphs with *fusion* and *fission* [86] when storing graph vertices and edges. Specifically, fusion organizes a set of neighboring low-degree vertices into a super-vertex, and fission splits a high-degree vertex into a set of sibling sub-vertices. The vertices and edges of the refactorized graphs are assigned to the nodes according to the proximity of the multi-dimensional tree topology.

To shorten the communication paths of BFS messages, we organized the CNs attached to the same HFR-E controller into one group (*i.e.*, communication domain). Owing to HFR-E's highly optimized on-chip routing mechanism, the overhead of intra-domain communication is much lower than that of inter-domain communication. This enables TuComm to perform topology-aware message aggregation to minimize the expected total number of hops in the BFS search.

Each communication domain has a responsible node (*i.e.*, monitor) which gathers messages from the same domain for transmission to other domains and receives messages from other domains for scattering within the same domain. The selection of monitors is performed as follows. First, monitors should contain heavy vertices (for locality). Second, place as many monitors as possible in the same HFR-E controller's routing table (for efficient mapping).

## 7 EXPERIMENTAL SETUP

### 7.1 Hardware Platforms & Graph Data

TuComm was tested on three HPC systems (Table 1) with different CPU architectures and interconnection components. We evaluated TuComm using six widely-used graph algorithms: BFS, SSSP, CC, BC, PR, and CDLP. Although our discussion primarily focuses on BFS, the other algorithms exhibited similar performance improvements. To thoroughly assess the scalability of TuComm, we used both synthetic and real-world datasets. The synthetic data was generated using the Graph500 tool, which creates large-scale graphs by taking two parameters: the graph factor ($m$) and edge factor ($n$). The generator produces a graph with $2^m$ vertices and $n \times 2^m$ edges. In our experiments, we varied the graph factor from 26 to 41 while keeping the default edge factor of 16, to create graphs of

**Table 2: Synthetic graph data used in our evaluation**

| scale[5] | #Vertices | #Edges | #Comp. Nodes |
|---|---|---|---|
| 26 | 64 M | 1 B | 1 |
| 27 | 128 M | 2 B | 2 |
| 28 | 256 M | 4 B | 4 |
| 29 | 512 M | 8 B | 8 |
| 30 | 1 B | 16 B | 16 |
| 32 | 4 B | 64 B | 64 |
| 34 | 16 B | 256 B | 256 |
| 36 | 64 B | 1 Tri | 1,024 |
| 37 | 128 B | 2 Tri | 2,048 |
| 38 | 256 B | 4 Tri | 4,096 |
| 41 | 2 Tri | 32 Tri | 79,024 |

**Table 3: Real graph data used in our evaluation**

| dataset | #Vertices | #Edges | #Comp. Nodes |
|---|---|---|---|
| com-Friendster [10] | 1.1 B | 91.8 B | 16 |
| clueweb12 [72] | 987 M | 42.6 B | 16 |
| twitter-2010 [34] | 4.2 M | 1.5 B | 16 |

different sizes. Details of the synthetic and real-world datasets used are listed in Table 2 and Table 3. These graphs were stored in the compressed sparse row (CSR) format to reduce memory usage.

### 7.2 Competing Baselines

We compare TuComm to two representative graph communication strategies: MST [19] and AML [27]. AML is a state-of-the-art message library for graph processing and it is built in the Graph500 implementation by default [27, 32]. MST is an optimized version of AML. We also compare TuComm with the representative partitioning schemes. In addition, we also compare TuComm with three state-of-the-art graph processing engines, namely GraphCube [20], GraphScope [14], and Gluon [11], using the engineer-tuned algorithm implementations provided by the frameworks.

## 8 EXPERIMENTAL RESULTS

### 8.1 Benchmarking Graph500

We have deployed TuComm to benchmark Graph500 BFS and SSSP on Tianhe-Exa. In our experiments, we used 79,024 computing nodes (1,264,384 cores) for BFS and 8,192 nodes (131,072 cores) for SSSP. We did not evaluate SSSP on a larger scale due to financial constraints. Our implementation and evaluation fully comply with the Graph500 specification.

The Graph500 ranking published in Nov. 2023, places Fugaku and Wuhan Supercomputer as the top performers for BFS and SSSP, respectively. However, TuComm on Tianhe-Exa successfully outperforms these top-ranking systems for both benchmarks, demonstrating the efficacy and competitiveness of TuComm.

TuComm achieved a throughput of 164,949 GTEPS using 1,264,384 processor cores for BFS, translating into more than 18.8% improvement over Fugaku's 138,867 GTEPS using 7.3 million cores (i.e., 5.8× more cores than TuComm). The advantages of TuComm come from the interconnection hierarchy-aware message aggregation and active buffer management. Our evaluation shows that message aggregation alone gives around 5× improvement over the standard,

---

[5]Edge factor of synthetic graphs is fixed at 16 [32].

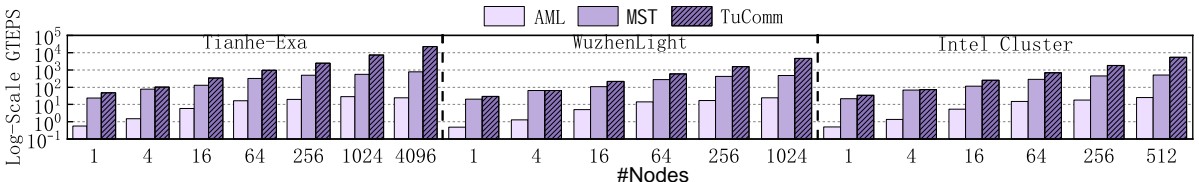

Figure 4: BFS throughput given by different communication strategies.

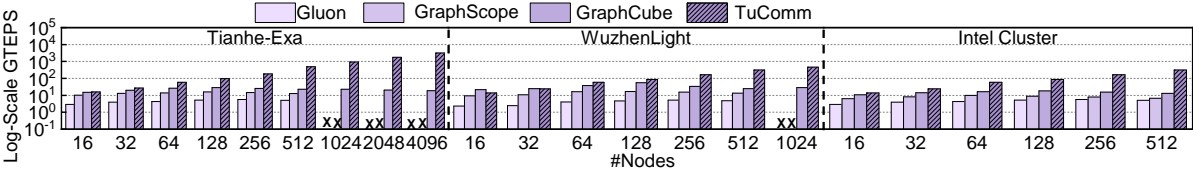

Figure 5: BFS throughput delivered by graph processing engines and TuComm.

parallel BFS implementation based on AML, and active buffer management gives a further 3× improvement over the standard BFS implementation.

For SSSP, TuComm achieved 23,021 GTEPS using 131,072 cores, representing a 50.1% improvement over the Wuhan Supercomputer's performance of 15,335.9 GTEPS. It is worth noting that Wuhan Supercomputer utilizes more cores (6,999,552) and has more memory per shared-memory node (2TB) compared to Tianhe-Exa (16GB per node). Wuhan Supercomputer is designed for data-analytic workloads with ample memory resources, enabling it to handle more vertices per node and optimize distributed graph processing challenges. In contrast, Tianhe-Exa has significantly less memory per shared-memory node and incurs more expensive communication overhead. Hence, TuComm's enhancements for SSSP are significant, given the considerable hardware advantages of the Wuhan Supercomputer.

## 8.2 Compare with Baseline Methods

Figure 4 compares TuComm to state-of-the-art graph communication strategies, namely AML and MST. Figure 5 compares TuComm against three graph processing engines: GraphScope, Gluon and GraphCube. The experiment used up to 4,096 Tianhe-Exa nodes to execute BFS. Some methods led to a runtime error and are marked as **X**, since they failed to exploit the discrepancy in hierarchical communication and incur huge communication overhead.

TuComm outperforms all baselines, particularly as the number of computing nodes increases. For instance, when processing a graph scale of 38 using 4,096 Tianhe-Exa nodes, TuComm delivers 22,490.17 GTEPS, 9.7× and 28.7× improvements over AML and MST, respectively. We also obtain similar results on SSSP, PR, CC, BC and CDLP, where TuComm respectively gives 27.2×, 29.1×, 25.6× and 19.7× throughput improvements over the best-performing baseline when using 4,096 Tianhe-Exa nodes.

## 8.3 Preprocessing Overhead

Before graph algorithms ingress, typical graph processing involves a *preprocessing* that performs tasks such as discarding isolated vertices, counting degrees, and sorting vertices by edge degrees. Figure 6 reports the time spent on the preprocessing. Generally, as

the size of the graph and the number of computing nodes increases, the preprocessing overhead also grows. However, we observe that TuComm has the lowest preprocessing overhead compared to other methods. In contrast, GraphScope, which requires significant *preprocessing* of the input graph, incurs 70.15× longer processing time than TuComm.

## 8.4 Communication Volume & Time

Since communication takes most of the overall time for large-scale graph processing, we compare the communication volume reduction and communication time of TuComm over AML for BFS on Tianhe-Exa.

In Figure 7(a), we can observe that TuComm is much better than that of the state-of-the-art AML and MST for communication reductions, which indicates that TuComm could trade cheap intra-domain communications for expensive inter-domain communications. We further examine the communication volume reductions varying nodes from 128 to 1,024, whose results are shown in Figure 7(b), where TuComm significantly outperforms other baselines. The advantage grows significantly as the number of computing nodes or graph size increases.

## 8.5 Scalability of TuComm

Figure 8 shows the scalability of TuComm against various state-of-the-art AML and MST running BFS on Tianhe-Exa. In contrast, prior solutions struggle to scale beyond 256 nodes because they overlook the communication variation within multi-level communication hierarchies. However, TuComm delivers higher throughput than baselines and scales well beyond 4,096 nodes.

## 8.6 Graph Operations on Real-world Data

We finally test TuComm on public datasets in Table 2 using 16 Tianhe-Exa nodes across two blades. Figure 9 compares TuComm with GraphScope, Gluon, and GraphCube, which offer engineer-optimized implementations for the test algorithms. TuComm consistently outperforms GraphScope, Gluon, and GraphCube in all test cases during the graph computation stage, achieving a speedup of up to 18.92×, 23.56× and 27.34× over GraphScope, Gluon, and GraphCube, respectively.

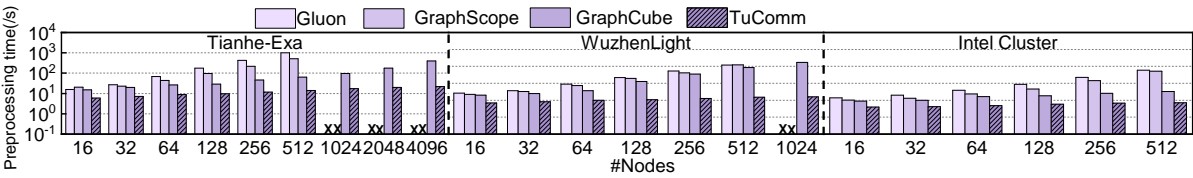

Figure 6: Graph preprocessing overhead (*lower-is-better*).

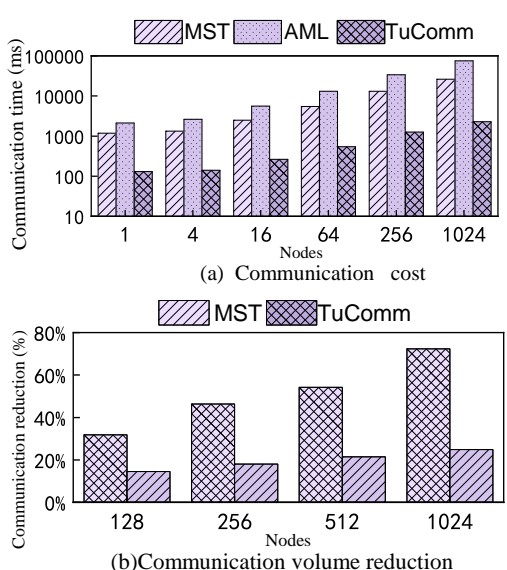

(a) Communication cost

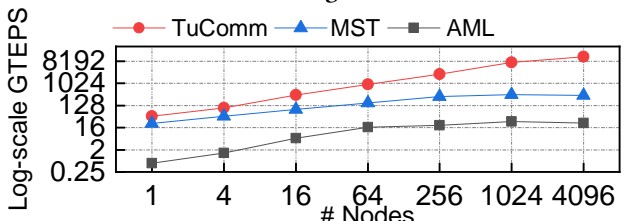

(b)Communication volume reduction

Figure 7: Communication time in (a) and (b) shows the communication volume reduction against AML.

Figure 8: The scalability of TuComm vs. MST and AML when running BFS on Tianhe-Exa.

## 9 RELATED WORK

Communication is of particular importance for training graph-based LLMs on HPC systems, like exascale supercomputers, where AML is the de facto standard communication library. AML-based communication libraries have been widely adopted to communication-intensive scenarios, such as parallel active message interface (PAMI[41]) and low-level applications programming interface (LAPI[64]) for IBM series supercomputers and K series computers[58], and MPI-3 RMA for TACC Stampede [44]. Hasanov *et al.* redesign the collective communication for operations of *Reduce* and *Allreduce* built in MPI, which effectively reduces the communication cost of clusters with a two-level hierarchy [31].

However, those MPI-based optimizations cannot adapt to graph processing on large-scale HPC systems like supercomputers, as the processing of graphs is quite different from traditional computation-intensive applications [20, 22, 48, 70, 71, 87].

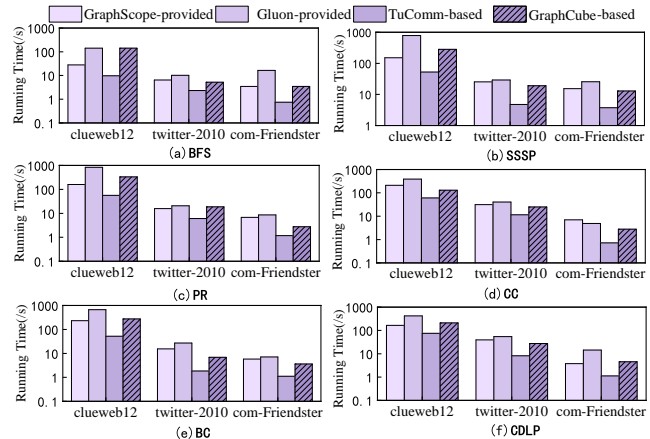

Figure 9: Performance comparison on real-world graphs.

To address this problem, the Graph500 community implements the AML for large-scale graph traversal. AML improves communication efficiency via per-node message aggregation, where each source node aggregates messages if they are sent to the same lowest-level target domain. However, when processing trillion-scale graphs on large-scale HPC systems with hierarchical communication domains, the communication cost of AML remains overwhelming and thus severely affects the graph processing performance. TRAM [76] is another communication library for communication-intensive applications of supercomputers, which routes messages along the dimensions of a virtual topology using intermediate relay nodes to dynamically combine the messages having the same destinations. Similar to AML, TRAM has no optimization for cross-domain communication. Unlike prior communication optimizations and message libraries, TuComm is presented to reduce cross-domain communications.

## 10 CONCLUSION

We have presented TuComm, a communication engine designed to accelerate training graph-based LLMs using hierarchical HPC systems. By modeling latency across the communication hierarchy, TuComm performs more aggressive message aggregation than AML to reduce cross-domain communication. Extensive evaluations of TuComm involved the Graph500 benchmark and fundamental graph operations across three renowned large-scale HPC systems, utilizing over 79K CNs and more than 1.2 million processor cores. The results demonstrate that TuComm consistently surpasses state-of-the-art baselines and other graph processing systems, highlighting its potential to significantly improve performance in distributed training graph-based LLMs.

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
