# OpenReview forum: "Communication Hierarchy-aware Graph Engine for Distributed Model Training"
_ACM.org/TheWebConf/2025/Conference — WWW 2025 Oral_

### Official Review · Reviewer_T6Q5 · 2024-11-15

**Novelty:** 5
**Technical Quality:** 6

**Review:**

This paper proposes TuComm, a framework for managing the communication for graph processing on high performance computer (HPC). TuComm observes that HPC usually has complex communication hierarchy, and the delays of high-level communication is much longer than low-level. TuComm merges the messages in same communication domain to send them to the same target domain via one message. The empirical results show that TuComm achieves higher processing speed and lower communication costs than the baselines.

Strong points

S1: The complex communication hierarchy of high-performance computer is explained clearly.

S2: The empirical results are good, showing that TuComm outperforms the baselines.

Weak points

W1: The design is rather simple, i.e., merging the messages. The implementation also seems straightforward.

W2: The connection between the motivation and solution is not clear. The motivation is that communication in high-level has much longer latency than low-level, while the solution is to merge the messages to the same target domains. The equations directly add the communication delays of different node pairs, this may not be reasonable because different communication links can work in parallel. Moreover, why merging helps to reduce communication delay? To reach its destination, a message needs to go through the communication hierarchy, and merging does not reduce the number of hops. Should not the reason be that merging reduces the costs of processing many small messages at high-level?

W3: The writing needs to be significantly improved to meet the standard of a top conference.

(1)	The paper mentions graph-based LLMs in many places but the experiments are all about traditional graph processing algorithms.

(2)	In all bar plots, the bars have the same color, which makes it very difficult to distinguish different baselines.

(3)	Why are background and preliminaries put into different sections? As a system-side paper, do you need to the formal definitions in Section 3.1? It is strange that experiment setting (Section 7) and experiment results (Section 8) are put into different sections. Section 8 has too many subsections, you may simplify classify them as main results and micro experiments.

(4)	I do not see the communication latency modeling in Section play any role in justifying the designs in Section 4.

(5)	Figure 3(b) does not show communication merging and hence fails to illustrate the main idea of TuComm.

(6)	The text in algorithm 1 is too small to read. Instead of using an algorithm, you may write the procedure as several steps with plain text.

Overall, I think the paper is not ready for publication and needs significant revision.

**Questions:**

See weakness

**Reviewer Confidence:**

3: The reviewer is confident but not certain that the evaluation is correct

**Scope:**

4: The work is relevant to the Web and to the track, and is of broad interest to the community

---

### Official Review · Reviewer_Nj7U · 2024-11-24

**Novelty:** 6
**Technical Quality:** 7

**Review:**

This work is about mitigating communication cost in distributed graph processing and is relevant to the track "Systems and infrastructure for Web, mobile and WoT". It proposes a better solution (TuComm) to mitigate the distributed communication cost than current distributed communication libraries like AML and MST.
The core idea is to aggregate/batch small messages within the same domain first rather than across domain first, then transmit them to the target domain in batch, so that the system can leverage more cheap intra-domain communication and less expensive inter-domain communication.
This work realizes some graph algorithms based on TuComm and implements experiments to compare TuComm with mainstream communication strategies and graph processing engines on large-scale HPC systems. All experiments show TuComm outperforms all baselines in metrics like GTEPS and communication time and has better scalability, so its technical claims are sufficiently supported.

**Questions:**

One question I got confused though is that it says "when processing a graph scale of 38 using 4,096 Tianhe-Exa nodes, TuComm delivers 22,490.17 GTEPS, 9.7× and 28.7× improvements over AML and MST", but the 9.7x and 28.7x seems not right according to Figure 4.

**Reviewer Confidence:**

3: The reviewer is confident but not certain that the evaluation is correct

**Scope:**

4: The work is relevant to the Web and to the track, and is of broad interest to the community

---

### Official Review · Reviewer_yyz6 · 2024-11-29

**Novelty:** 4
**Technical Quality:** 4

**Review:**

This paper proposed a new communication engine used in distributed training of graph-based LLMs. The introduction and motivation sections clearly define the challenge posed by hierarchical communication domains in large-scale high-performance computing (HPC) systems. The proposed TuComm framework is grounded in solid theoretical models and is implemented with comprehensive technical detail. The experimental setup is comprehensive, and the results clearly demonstrate the superiority of the proposed method in terms of performance, scalability, and communication volume.
P1: The paper is written in a largely clear manner that makes it easy to follow.
P2: Comprehensive experiments demonstrating the performance gains of TuComm.
P3: Experimental improvements over baselines are impressive.
C1: Some details are not adequately considered, such as the input not being reflected in the pseudocode.
C2: The logic between some of the subsections in the paper is inconsistent.
C3: The methodology, while detailed, could have benefited from a clearer explanation of the underlying assumptions in communication cost modeling.

**Questions:**

Q1: The input of Algorithm 2 does not seem to be reflected in the subsequent execution logic. Could this be a misunderstanding, or is there an omission or inconsistency in the description of the algorithm's workflow?
Q2: The paper focuses on network communication scenarios between RC and CN but does not address load balancing between different CN computations. How does the proposed approach handle scenarios with uneven computational loads across CNs, and are there plans to incorporate load balancing mechanisms to improve scalability and efficiency in such cases?
Q3: The paper does not discuss potential trade-offs in terms of memory usage, energy consumption, or fault tolerance. Given that the system involves a massive computing infrastructure with tens of thousands of nodes, why is fault tolerance not considered? Addressing fault tolerance is crucial in such large-scale systems to ensure reliability and robustness in the face of inevitable node or network failures.

**Reviewer Confidence:**

3: The reviewer is confident but not certain that the evaluation is correct

**Scope:**

3: The work is somewhat relevant to the Web and to the track, and is of narrow interest to a sub-community

---

### Official Review · Reviewer_MbCZ · 2024-12-01

**Novelty:** 7
**Technical Quality:** 7

**Review:**

This paper addresses a critical and challenging problem: performing graph computations over large-scale graphs in distributed computing systems consisting of thousands of nodes. The authors propose a system, named TuComm, which introduces a dynamic approach to aggregating and transferring messages. The core contribution of TuComm is leveraging interconnection network topology to enable domain-level (as opposed to node-level) message aggregation. In this approach, messages destined for the same target domain are aggregated within their source domain before transmission, thereby optimizing communication efficiency.

Strong Points:
S1: The paper tackles a significant and challenging problem in distributed graph computation.

S2: The system is tested on graphs and distributed computing setups that are significantly larger in scale than those in most existing studies. The evaluation is robust, utilizing the widely recognized Graph 500 benchmark to validate the results.

**Questions:**

W1: While the authors claim that TuComm can support distributed model training, the paper does not provide verification using traditional graph machine learning tasks, such as graph neural networks.

W2: The proposed hierarchy-aware message aggregation mechanism could benefit from a more formal and detailed discussion.

**Reviewer Confidence:**

4: The reviewer is certain that the evaluation is correct and very familiar with the relevant literature

**Scope:**

4: The work is relevant to the Web and to the track, and is of broad interest to the community